# High Behavioral Reactivity to Novelty as a Susceptibility Factor for Memory and Anxiety Disorders in Streptozotocin-Induced Neuroinflammation as a Rat Model of Alzheimer’s Disease

**DOI:** 10.3390/ijms252111562

**Published:** 2024-10-28

**Authors:** Joanna Dunacka, Grzegorz Świątek, Danuta Wrona

**Affiliations:** Department of Animal and Human Physiology, Faculty of Biology, University of Gdansk, 59 Wita Stwosza Str., 80-308 Gdansk, Poland; joanna.dunacka@ug.edu.pl (J.D.); grzegorz.swiatek@phdstud.ug.edu.pl (G.Ś.)

**Keywords:** neuroinflammation, rat Alzheimer’s disease model, high and low responders to novelty, spatial memory and anxiety disorders, streptozotocin, white and illuminated open field, elevated plus maze, Morris water maze, β-amyloid, cytokines

## Abstract

Individual differences in responsiveness to environmental factors, including stress reactivity and anxiety levels, which differ between high (HR) and low (LR) responders to novelty, might be risk factors for development of memory and anxiety disorders in sporadic Alzheimer’s disease (sAD). In the present study, we investigated whether behavioral characteristics of the HR and LR rats, influence the progression of sAD (neuroinflammation, β-amyloid peptide, behavioral activity related to memory (Morris water maze) and anxiety (elevated plus maze, white and illuminated open field test) in streptozotocin (STZ)-induced neuroinflammation as a model of early pathophysiological alterations in sAD. Early (45 days) in disease progression, there was a more severe impairment of reference memory and higher levels of anxiety in HRs compared with LRs. Behavioral depression in HRs was associated with higher expression of β-amyloid deposits, particularly in the NAcS, and activation of microglia (CD68^+^ cells) in the hypothalamus, as opposed to less inflammation in the hippocampus, particularly in CA1, compared with LRs in late (90 days) sAD progression. Our findings suggest that rats with higher behavioral activity and increased responsivity to stressors show more rapid progression of disease and anxiety disorders compared with low responders to novelty in the STZ-induced sAD model.

## 1. Introduction

Alzheimer’s disease (AD) is characterized by multiple cognitive deficits as a result of neuronal and synaptic loss [1]. The pathophysiological characteristics include extracellular formation of senile plaques consisting of β-amyloid (A*β*) peptide, intracellular accumulation of aggregates of hyperphosphorylated tau protein, neuronal and synaptic loss, proliferation of astrocytes and activation of microglia [2,3,4]. Although the exact cause of AD remains poorly understood, neuropathological studies have supported early involvement of neuroinflammation in AD by demonstrating the accumulation of activated microglia and inflammatory mediators in the cerebral neocortex at a low stage for AD pathology [5,6,7]. Microglia are involved in AD pathogenesis through the release of inflammatory mediators known to contribute to the production and accumulation of A*β*. On the other hand, A*β* itself, an inducer of microglia activation and neuroinflammation, has been considered as a primary and unifying factor in the development of AD. A vicious cycle of inflammation arises between A*β* accumulation, activated microglia, and microglial inflammatory mediators, which increase A*β* deposition and neuroinflammation.

The risk of AD increases significantly with age. Endothelial dysfunction and inflammaging are the most frequent impairment characterizing aging patients [8,9,10]. AD has multiple etiological factors, including genetics or environmental conditions, which also play a role in the development of individual differences in stress sensitivity and disease susceptibility [11]. There is considerable heterogeneity between individuals in their response to these environmental factors [12]. Differences in behavioral characteristics might be due to differences in baseline anxiety or stress levels among patients or between animals used in studies. Individual differences in stress reactivity measured as behavioral response to novelty (high (HR) or low (LR) responders to novelty) are closely related to differences in anxiety and susceptibility/resistance to stress [13].

HR rats characterize increased responsivity (behavioral, hormonal, and neurochemical) to stressors, including higher and prolonged secretion of corticosterone (CORT) in response to a novelty stress challenge [14,15,16]. The effect of the level of spontaneous locomotor activity on the risk of AD has not yet been investigated. Therefore, in the present study, we tested the hypothesis that the animal’s behavioral characteristics (HRs vs. LRs), and thus baseline anxiety/stress levels, are risk factors for the development of memory impairment and anxiety disorders in an animal model of AD, and that these factors determine differences in the neuroimmune correlates of disease between individuals. Since most cases of AD are sporadic rather than hereditary [17,18], we used rats subjected to intracerebroventricular (ICV) injections of streptozotocin (STZ), which proved to be the best animal model of early pathophysiological changes in sporadic AD (sAD) [19]. ICV-STZ-induced neuroinflammation is a sporadic AD model based on brain resistance to insulin [19,20,21,22], which mimics many physiopathological aspects of sAD in human, such as memory impairment, changes of glucose metabolism, neuroinflammation, oxidative stress and phosphorylation of tau protein [23,24,25]. Therefore, we examined behavioral activity related to memory processes (Morris water maze (MWM), anxiety (elevated plus maze; EPM; white and illuminated open field test; OF), central (CD68^+^ microglia cell number) and peripheral (T lymphocyte CD4^+^/TCD8^+^ number, interleukin (IL)-6/IL-10 concentration) functions of the immune system playing a key role in cognitive and memory processes. To test for differences in the rate of AD progression between HR and LR rats, we examined these parameters before (baseline) ICV-STZ-induced neuroinflammation as a model of early pathophysiological alterations in sAD and at early (45 days) and late (90 days) of disease progression.

## 2. Results

### 2.1. Behavioural Results

#### 2.1.1. Novelty Test

The median value of the locomotor activity score (based on horizontal movements) for the whole group (n = 31) tested was 4607. The lowest value within the whole group was 1320, while the highest value was 6794. The animals with an activity score above the median were designated as HRs (n = 16, 5565 ± 574, mean ± SD), while those with a motility score below the median were designated as LRs (n = 14, 2275 ± 854 mean ± SD). Rat with the median activity score was rejected from the further experiments.

#### 2.1.2. Fear/Anxiety, Spatial Memory, and Learning in the Elevated Plus-Maze Test (EPM)

Behavioral activity assessed in the EPM test measured as time spent in the open arms, time spent in the closed arms and in the center of the maze, as well as and the number of entries into the open arms, closed arms, and into the center during the test and 1 h after test (re-test) are presented in Figure 1 and Appendix A. As compared to the baseline and control VEH group, rats with the STZ-induced AD model spent a significantly less time in the open arms and center of the maze, whereas they spent more time in the closed arms on the 45th and 90th post-injection days (Appendix A). There was a significant difference (*p* ≤ 0.05) in both time spent and the number of entries into the open arms, center, and closed arms of the EPM between HRs and LRs within the STZ group. As opposed to the baseline (HR vs. LR, *p* ≤ 0.01), the number of entries into the open arms and time spent in the center were decreased (*p* ≤ 0.05) in the HRs as compared to LRs on the 45th and 90th days after AD induction (Figure 1, Appendix A). Moreover, significantly decreased time spent in the center (*p* ≤ 0.001) of the EPM 45 days after STZ injections, whereas increased time spent in the open arms (*p* ≤ 0.05) and center (*p* < 0.01) on the 90th post-injection day during the re-test rather than test in the STZ group was observed (Appendix A). However, all measured behavioral parameters returned to the baseline values in re-test on the 90th day after STZ injection in LRs only.

#### 2.1.3. Behavioral Activity and Fear/Anxiety in the White and Illuminated Open Field Test (OF)

Behavioral activity in the white and illuminated open field test measured as an exploration (a number of crossed lines), rearing, grooming, time of freezing, defecation, and miction frequency at the baseline and on the 45th and 90th days after STZ or VEH injection are presented in Figure 2 and Table 1. The significantly higher amounts of exploration (*p* ≤ 0.001), rearing (*p* ≤ 0.01), miction (*p* ≤ 0.05), and defecation (*p* ≤ 0.05) in the STZ group as compared to the control VEH on the 45th post-injection day was observed (Figure 2, Table 1). As shown in Figure 2, time of freezing was increased (*p* ≤ 0.05) within the STZ group in comparison with the control animals both on the 45th and 90th post-injection days. As compared to the baseline, in the STZ group, the decreased behavioral activity including an exploration (*p* ≤ 0.001), entries into the center (*p* ≤ 0.01), and time spent in the center (*p* ≤ 0.01) of the OF maze and miction (*p* ≤ 0.001), whereas increased time spent in the periphery (*p* ≤ 0.01) was observed 45 and 90 days following injections. Moreover, decreases in rearing (*p* ≤ 0.001) and grooming (*p* ≤ 0.05) in the STZ group in comparison with the baseline values were observed on the 90th post-injection day only.

There were significant differences in the number of crossed lines (exploration) and time of freezing between the HR and LR animals within the STZ and VEH groups (Figure 2). Increased exploration number (*p* ≤ 0.05) in the STZ HRs and VEH HRs rather than STZ LRs or VEH LRs were observed 45 days after injections. Moreover, the higher amounts of grooming (*p* ≤ 0.05) in the HRs within the STZ group as opposed to the lower amounts of grooming (*p* ≤ 0.05) in the VEH HRs rather than the LRs of the respective groups were noticed. In contrast to the HR vs. LR differences at the baseline (*p* ≤ 0.01) and in the VEH animals 90 days after injections (*p* ≤ 0.01), on the 45th post-injection day, the HRs within the STZ group had longer time of freezing compared to LRs (*p* ≤ 0.05; Figure 2).

#### 2.1.4. Behavioral Activity Associated with Memory Impairments in the Morris Water Maze Test (MWM)

Behavioral activity in the Morris water maze test measured as latency to reach the platform, total distance swum, and time spent in critical quadrant at the baseline and on the 45th and 90th day after STZ or VEH injection are presented in Figure 3 (on the day of probe test with removed platform) and Table 2 (Reference memory during 1–3 consecutive days of MWM and working memory during Trial 1–4 of one day). As shown in Figure 3a, on the day of probe the latency to reach the platform was significantly (*p* ≤ 0.05) longer in the STZ group as compared to the VEH animals on the 45 and 90 post-injection days. There was significantly (*p* ≤ 0.01) longer latency to reach the platform both in the STZ and VEH groups on the 90th as compared to 45th post-injection day and baseline (*p* ≤ 0.001). On the 90th post-injection day, a significantly lower percentage of time spent in the critical quadrant as compared to the control VEH animals (*p* ≤ 0.05) and baseline (*p* ≤ 0.01) in the STZ animals was observed.

As shown in Figure 3b, significant HR vs. LR differences within the STZ animals in all measured parameters on the probe test were observed. A significantly longer the latency to reach the platform at baseline and on the 45th post-lesion day (*p* ≤ 0.05) in HRs rather than LRs within the STZ animals were observed. In HRs within the STZ group, lower percentage of time spent in the critical quadrant and total distance swum (*p* ≤ 0.05) were noticed on the 90th day after AD model induction as compared to the LRs and to the control VEH HR animals (*p* ≤ 0.05).

Working memory measured in the MWM that is indicated as the latency to reach the platform during four trials (Trials 1–4) per one day is presented in Table 2a. As compared to the controls, a significantly (*p* ≤ 0.001) longer latency to reach the platform from Trial 1 to Trial 4 on the 45th and 90th post-injection days in the STZ animals was observed. The HR vs. LR difference in the latency to reach the platform within the STZ animals was observed during Trial 2 (*p* ≤ 0.001), Trial 3 (*p* ≤ 0.001) and Trial 4 (*p* ≤ 0.05) on the 90th day after AD induction, with lower latency in the HRs rather than LRs.

Reference memory measured as the latency to reach the platform during three consecutive days of MWM (Days 1–3; Table 2b) was significantly impaired, as indicated by the longer latency to reach the platform within the STZ animals as compared to the control group, both on the 45th and 90th post-injection day. Moreover, the total distance swum was significantly longer (*p* ≤ 0.001) in the STZ animals rather than the control VEH group on the 45th and 90th day after AD induction. There were significant differences in the latency to reach the platform and total distance swum (*p* ≤ 0.05) between the HR and LR animals within the STZ group, with longer latency both on the 45th and 90th days after STZ injections and shorter total distance swum (*p* ≤ 0.05) on the 45th post-injection day in the HR compared to LR rats.

### 2.2. Microglia Activated Cells (CD68^+^ Cells) and Amyloid β (Aβ_40–42_) Deposits in the Hippocampal Areas (CA1, CA2, CA3, DG) and Nucleus Accumbens (NAcC and NAcS)

As shown in the Figure 4a, a significantly higher number of CD68^+^ cells in the STZ (*p* ≤ 0.001) rather than VEH group as well as HRs and LRs of the STZ group and their behavioral counterparts in the control VEH animals in all hippocampal areas (CA1, CA2, CA3, *p* ≤ 0.01; DG, *p* ≤ 0.05) and the nucleus accumbens (NAcS: HR, LR *p* ≤ 0.05; NAcC: HR *p* ≤ 0.05, LR *p* ≤ 0.01) was observed on the 90 day after STZ injection. There were no significant differences in the CD68^+^ cell number in the hippocampal areas and NAcS and NAcC observed within the VEH group. Additionally, as compared to the VEH group, a significantly higher number of activated microglia cells in the limbic structures (prefrontal cortex: *p* ≤ 0.001), corpus callosum: *p* ≤ 0.001, caudate putamen: *p* ≤ 0.01, medial septal nucleus: *p* ≤ 0.001), preoptic area (medial and lateral preoptic nuclei: *p* ≤ 0.001, supraoptic nucleus: *p* ≤ 0.05) and hypothalamic nuclei (arcuate: *p* ≤ 0.01, dorsomedial: *p* ≤ 0.001, lateral: *p* ≤ 0.001, paraventricular: *p* ≤ 0.001, ventromedial: *p* ≤ 0.001) were observed (Table 3). Moreover, both the HR and LR animals within the STZ group had a significantly higher numbers of CD68^+^ microglia cells in all studied brain areas, except for the arcuate and supraoptic nuclei (Table 3) rather than the VEH animals. There was a significant HR vs. LR difference (*p* ≤ 0.05) in the number of CD68^+^ cells within the STZ group with a lower cell number in the CA1 part of the hippocampus (Figure 4a and Figure 5A), whereas it was a higher microglia cell activation in the dorsomedial hypothalamic nucleus in HRs rather than LRs (Table 3). Furthermore, a significant positive correlation (R = 0.92, *p* ≤ 0.01) between the number of CD68^+^ microglia cells in the CA1 part of the hippocampus and the IL-6 plasma concentration was found in STZHRs.

As compared to the VEH animals, amyloid β (Aβ_40–42_) deposits were observed in the hippocampal areas and nucleus accumbens in the STZ-injected animals in the 90 days after an induction of the AD model (Figure 4b). There were no Aβ_40–42_ deposits noted in the studied brain areas within the VEH animals. The significantly higher number of Aβ_40–42_ peptides in the hippocampal areas (CA1, CA2, CA3, DG) and nucleus accumbens (shell and core) were observed in the STZ animals and both STZHRs and STZLRs groups (Figure 4b and Figure 5B). As shown in Figure 4b and Figure 5B, within the STZ group, the higher level of Aβ_40–42_ was observed in HRs rather than LRs in the hippocampal areas and nucleus accumbens. However, the significance of difference in Aβ_40–42_ number between STZHRs and STZLRs was reached in NAcS only (*p* ≤ 0.01).

### 2.3. Peripheral Blood Leukocyte and Lymphocyte and Their Subpopulation Number

There were no significant differences in the peripheral blood leukocyte as well as the total and percentage number of lymphocytes between the STZ and VEH groups and their behavioral counterparts (HR vs. LR) on the 45th and 90th post-injection days (Appendix A). However, significant differences in the number of peripheral blood lymphocyte subpopulations (T, B, NK), granulocytes, and monocytes between STZ and VEH as well as between HRs and LRs within the STZ group were noticed. As compared to the controls, in the STZ group, the total number of TCD3^+^ lymphocytes (*p* ≤ 0.01), especially in STZLRs and TCD4^+^ subpopulation of T lymphocytes (Figure 6), the total (*p* ≤ 0.05) and percentage (*p* ≤ 0.01) number of monocytes (Appendix A) increased on the 45th day after injection. Ninety days after the STZ injections, a significantly (*p* ≤ 0.01) lower NK (CD161a^+^) cell number, in comparison with the controls, was observed (Figure 6).

Forty-five days after the STZ injections, a higher total number of B (CD45RA^+^) lymphocytes (*p* ≤ 0.05), concomitantly with a lower number of NK cells (CD161a^+^) (*p* ≤ 0.05; Figure 6), granulocytes, and monocytes (*p* ≤ 0.05; Appendix A), in HR rather than LR animals was noticed. As opposed to the control VEH animals, there was a significantly lower (*p* ≤ 0.05) total number of B (CD45RA^+^) lymphocytes in HRs as compared to LRs within the STZ group on the 90th post-injection day.

### 2.4. Relative Thymus and Spleen Weights

There was no significant difference in the relative thymus and spleen weights between the STZ (thymus: 31.16 ± 9.31 mg/100 g b.w., spleen: 183.82 ± 12.27 mg/100 g b.w.) and VEH (thymus: 33.56 ± 15.40 mg/100 g b.w., spleen: 175.47 ± 17.81 mg/100 g b.w.) animals on the 90th post-injection day. However, within the STZ group, significantly (*p* ≤ 0.05) lower relative thymus and spleen weights in HRs (thymus: 23.54 ± 6.6 mg/100 g b.w., spleen: 172.95 ± 10.85 mg/100 g b.w.) were found compared to LRs (thymus: 38.53 ± 5.28 mg/100 g b.w., spleen: 193.73 ± 2.66 mg/100 g b.w.). There was no significant difference in the relative thymus and spleen weights between HRs and LRs within the VEH group.

### 2.5. Plasma Interleukin-6 (IL-6)/Interleukin-10 (IL-10) and Corticosterone (CORT) Concentration

There were no significant differences in plasma IL-6 concentrations between the STZ and VEH animals and their behavioral counterparts (HR vs. LR) on the 90th day after injections (Appendix A). On the other hand, an increased level of IL-10 in the STZ group, in comparison with the controls (*p* ≤ 0.05), was observed. Moreover, a significantly higher (*p* ≤ 0.05) plasma CORT concentration in the STZ animals (209.17 ± 87.16 ng/mL), rather than the control group (147.90 ± 56.16 ng/mL), especially in the STZLRs (225.74 ± 102.71 ng/mL, *p* ≤ 0.01), was noticed on the 90th day after STZ injection. Within the VEH animals, the higher plasma CORT concentrations in VEHHRs (167.20 ± 65.51 ng/mL) rather than VEHLRs (123.79 ± 30.97 ng/mL) were noticed. However, no significant differences in this parameter between HRs and LRs within both the STZ and VEH groups were observed.

### 2.6. Hematological Parameters

There were significant differences in some hematological parameters, including red blood cell (RBC) number, hemoglobin (HGB) concentration, hematocrit (HCT), and mean hemoglobin concentration in the red blood cell (MCHC) between the STZ and VEH groups and their behavioral counterparts (HR vs. LR) within the STZ animals (Appendix A). As compared to the controls, a significantly higher number of RBC in the STZ group on the 45th (*p* ≤ 0.01) post-injection day was noticed. HGB concentration decreased 45 days (*p* ≤ 0.05) after the STZ injections whereas it increased (*p* ≤ 0.01) concomitantly with the MCHC on the 90th post-injection day, as compared to the controls. Moreover, significant HR vs. LR differences in these parameters were noticed (Appendix A), with lower RBC number (*p* ≤ 0.05), HGB concentration (*p* ≤ 0.05), and HCT (*p* ≤ 0.01), whereas higher MCHC (*p* ≤ 0.05) in HRs rather than LRs within the STZ animals.

## 3. Discussion

The current findings, to the best of our knowledge, are the first to show an interaction between individual behavioral characteristics and stress reactivity of an animal, measured as the differences in a spontaneous locomotor response to novelty (HRs vs. LRs), on the behavioral activity related to memory and fear/anxiety disorders, and neuroimmunological responses in rats with the streptozotocin-induced neuroinflammation model of sAD. In the present report, we provide the first evidence that rats with higher behavioral activity (high responders to novelty) and increased sensitivity to stressors show faster progression of sAD and disease-related anxiety disorders. An inherent limitation of our research is using the rat model to study sAD in humans and thus the translational value of animal-based results. However, we believe that the results of our studies may provide diagnostic and prognostic support and inspire new approaches to prevent or delay dementia in humans by modifying environmental risk factors throughout the life course.

The differences in the behavioral responses to novelty are known to be related to variations in the central dopaminergic and serotonergic pathway [26,27,28,29,30]. The HRs show higher dopaminergic activity in the nucleus accumbens (NAc) or dorsal striatum but lower dopaminergic activity in the prefrontal cortex at baseline than LR animals [29]. In the present study, at a late stage of the disease (90 days after induction of the AD model), greater neuroinflammation (increased CD68^+^ microglia cell number) and elevated β-amyloid levels in the prefrontal cortex were associated with lower microglia activation but higher β-amyloid numbers in the NAc in HR rats compared to LRs. Thus, our immunohistochemical and behavioral data suggest a correlation between brain dopamine, microglia immune response, and β-amyloid accumulation and thus impairment of cognitive processes in the ICV-STZ-induced model of AD. The correlation between brain dopamine and immune system response has been previously described [31]. In fact, we too have previously observed a correlation between spontaneous locomotor activity (HR vs. LR) and the level of immune system response in rats [32,33,34,35].

Regarding hippocampal-dependent spatial memory impairment, especially reference memory, were more pronounced in HRs compared to LRs, and indicated as a longer latency to reach the platform during Days 1–3 and shorter time swum in the critical quadrant during the probe test (without the platform) in Morris water maze test on day 45 after the ICV-STZ injection. These effects suggest an impairment of hippocampal function at the early stage of AD development in HRs. In fact, we observed larger β-amyloid deposits in the hippocampal CA1, CA2, CA3, and dentate gyrus, in STZHRs as compared to STZLRs. Thus, it is possible that our findings of greater memory impairment in HRs in the AD model may be due to damage of the hippocampal projections, in particular from the CA1, which terminate in the prefrontal cortex or NAc shell. Besides the hypothalamus, the hippocampus, prefrontal cortex and NAc are involved in behavior associated with stress regulation, anxiety and memory processes. Other authors reported that the dorsal hippocampus is critical for learning and memory whereas the ventral hippocampus is associated with emotion-related behavior, especially those that are associated with anxiety [36,37,38]. A strong connectivity between the ventral CA1 and medial prefrontal cortex has been reported to be related to anxiety [38,39,40,41]. The input of the ventral CA1 towards NAc also impacts anxiogenic behavior [42,43]. Furthermore, the prefrontal infralimbic cortex, necessary for fear extinction, projects preferentially to the shell subregion of the NAc [44,45].

Since reduced behavioral activity at baseline in LR rats was associated with lower glucocorticoid-induced dopamine release than in HR animals [26,27,28,46] and dopamine-dependent locomotion [47,48], we would expect that dopamine deficiency might be involved in depressed behavioral activity in tests of anxiety and/or memory impairment in LR animals after the STZ injections. Surprisingly, in contrast to the baseline, we observed behavioral depression associated with anxiety behavior, as measured by specific suppression of open-arm and center exploration in the elevated plus maze test on Days 45 and 90 after the AD induction and prolonged freezing time on Day 45 after the STZ injections in the white and illuminated open field test in HR animals. Elevated levels of anxiety and behavioral depression suggest that dopamine levels in the brains of the HR rats decline during disease progression, particularly in NAc. Several studies have implicated dopamine actions in NAc in conditioned fear. For example, dopamine activity in the NAc is implicated in the blocking of learned fear responses [49]. The NAc, a critical node for limbic motor integration, is a heterogenous structure divided into core (NAcC) and shell (NAcS) subregions which participate in emotional learning [44,50], including specific contribution of the NAcC and NAcS in regulating both fear expression and fear extinction [51,52,53]. The NAcC is specifically active only in response to an associative fear cue during an expression test. In contrast, the NAcS is specifically active during fear extinction [53]. Our evidence for increased behavior related to anxiety and memory disorders in HRs in the course of AD here is supported by additional immunohistochemical evidence that significantly more β-amyloid counts in NAcS in HRs compared to their absence in LRs were observed (Figure 4b and Figure 5B). These findings suggest that β-amyloid-induced damage to the NAcS and thus a reduction in dopamine levels in this part of NAc disrupted the consolidation of extinction of fear response in HRs.

On the other hand, HRs vs. LRs differences in response to novelty stress may be related to individual stress reactivity and differences in the hypothalamic—pituitary—adrenal (HPA) axis activity and glucocorticoid effects on memory processes [14,15,16,54]. HR animals have a higher and prolonged secretion of CORT in response to a novelty stress challenge and thus are considered as more susceptible to stress even though they have a lower baseline anxiety level [14,15,16]. They also exhibit a higher sensitivity to the behavioral and dopaminergic effects of glucocorticoids [29,46,55,56]. Therefore, we wanted to determine whether HRs and LRs in the AD model differ in HPA axis activity, as indicated by plasma CORT levels. In fact, at Day 90 after disease induction, elevated plasma CORT levels in the STZ group compared to the controls were due to increased hormone levels in LRs but not in HRs (Appendix A). It should be noted, however, that the higher stressor susceptibility and HPA axis response, and thus the prolonged elevated CORT levels in HRs early in the course of the disease, may explain not only the memory impairment but also why we did not observe significant differences in CORT levels between HRs and LRs in the STZ group 90 days post-injection. Furthermore, the elevated CORT levels in HRs may be responsible for peripheral suppression of their innate immunity, as indicated by the reduced total number of NK cells, monocytes and granulocytes and the lower relative thymus and spleen weight compared to LRs. It is possible that this effect is due to the greater sensitivity of immune cells and lymphoid organs in HR rats to such stress hormones as CORT than in LRs [15,55] and/or to the CORT-induced redistribution of peripheral immune cells to the other lymphoid organs [46]. Other CORT-induced peripheral immunomodulatory effects in the STZ rats in this study might have resulted from a shift of peripheral Th1 to Th2 lymphocytes and thus from a pro-inflammatory to an anti-inflammatory response in the late development of the disease. Indeed, we found an increased total number of TCD3^+^ lymphocytes in the peripheral blood, particularly the TCD4^+^ subpopulation, and higher plasma levels of anti-inflammatory IL-10 but not pro-inflammatory IL-6 in rats with the ICV-STZ model of AD at Day 90 post-injection supporting such an anti-inflammatory effect. Other authors [57] reported that relative suppression of pro-inflammatory IL-6 production was associated with impairment of memory retrieval of emotional, but not neutral words. According to the authors, glucocorticoid sensitivity and thus impairment of emotional memory retrieval is correlated with suppression of IL-6 cytokine production. It should be noted, however, that we failed to show significant differences in these peripheral blood immunological parameters, including plasma IL-6 concentration, between HRs and LRs within the STZ animals. Interestingly, we also found lower values for several peripheral hematological parameters, including total red blood cell (RBC) count, hemoglobin concentration (HGB), hematocrit (HCT) and mean cellular hemoglobin concentration (MCHC) in HRs but not LRs on day 45 after the ICV-STZ injection. Regarding a decrease in the RBC number in the ICV-STZ model of AD, the available data suggest that plasma β-amyloid peptides bind to ageing erythrocytes, implying a pathogenic role of erythrocyte amyloid complex in AD. This complex induces changes in the red blood cell morphology, adhesion to endothelium and influences vessel activity [58]. In the present study, the higher number of β-amyloid after the ICV-STZ injections observed in the HR animals, and thus the increased number of β-amyloid complexes with erythrocytes, may explain the reduced hematologic parameters and indicate the deterioration of red blood cell system function in these animals compared to LR rats with the AD model.

In addition to the effect of differences in animal behavioral reactivity on neuroinflammation, cognitive performance and anxiety level observed in the sAD model, the influence of other factors or confounding factors cannot be excluded. Other authors [59] found that the forced swim test induced markedly divergent hippocampal gene expression, as indicated by activation of the serotonergic system and neurogenesis in LR rats, whereas it increased levels of the pro-inflammatory enzyme indoleamine 2,3-dioxygenase and apoptotic mechanisms in HR animals despite their comparable depressive-like status. According to the authors, the hippocampus of HR and LR rats undergoes distinct transcriptional remodeling in response to the same stressor. Both in rats and humans, much of the variability of stress reactivity reflects individual differences in behaviorally evoked sympathetic nervous system activation. However, various degrees of inconsistency for risk factors exist between observational studies and clinical trials. Genetically, longer educational attainment has been well-established as associated with lower risk for AD, whereas other risk factors, including blood pressure, lipid traits, body mass index, smoking and alcohol consumption have shown inconclusive associations with AD. Based on genomic study [60], genetically determined modifiable risk factors, such as high-density lipoprotein cholesterol concentration and high systolic blood pressure, were associated with an increased risk of AD. Therefore, thoroughly unfolding the genomic background for association between modifiable risk factors and dementia might help to develop future effective preventive and therapeutic approaches.

## 4. Materials and Methods

### 4.1. Animals

All animal experiments were carried out in accordance with the European Communities Council Directive of 24 November1986 (86/609/EEC), and under the authority of the Local Ethical Committee for the Care and Use of Laboratory Animals at University of Technology in Bydgoszcz, Poland (No. 60/2017).

Four-month-old rats (n = 31) weighing 300 ± 80 g (Tri-City Central Animal Laboratory, Research and Service Centre of the Medical University of Gdansk, Poland). For the duration of the experiment, rats were housed separately in polycarbonate cages (20 cm width, 37 cm length, 18 cm height) on a 12 h light/dark cycle (lights on at 06:00) in an air-conditioned, constant-temperature (22 ± 2 °C) room could visually observe other subjects and were indirectly exposed to other subjects’ cage odors. Water and food were available ad libitum. Animals were allowed to adapt to the laboratory conditions for one week before the beginning of the experimental procedure. Then, rats were subjected to daily handling for about 5 min each day for two weeks in order to minimize stress caused by the procedure.

### 4.2. Experimental Procedure

#### 4.2.1. Novelty Test (NOV)

The novelty test was performed according to a method that we have described previously [32,33]. All of the rats were placed in the actometer (Opto Varimex Minor-Columbus, OH, USA) for 2 h (4–6 P.M. according to [13], where their locomotor activity was automatically recorded. Each of three photocell apparatus was constructed of clear plexiglass, measured 43 × 43 × 20 cm, and was equipped with 15 photocells. For each rat, the number of horizontal, vertical, and ambulatory plane photocell counts, accumulated for 10 min, were measured. The number of horizontal plane photocell counts accumulated over 2 h was used as an index of individual locomotor response to the new environment. Based on the median of activity score [13], rats were divided into high (HR) or low (LR) responders to novelty. Subsequently, rats from each activity group were randomly divided into STZ (n = 16; HR = 8, LR = 8) or VEH (n = 14; HR = 7, LR = 7) group. After being assigned to experimental groups, the animals were subjected to further procedures according to the diagram presented in Figure 7.

#### 4.2.2. Intracerebroventricular (ICV) Injections of Streptozotocin (STZ)—sAD Induction

ICV injections of streptozotocin (STZ) or vehicle (VEH: 0.02M citrate buffer pH 4.5) were performed according to our previous study with some modifications [61]. Briefly, rats under 2.5% isoflurane (airflow: 0.5 L per min) anesthesia using an isoflurane pump (Bitmos OXY 6000, Bitmos GmbH Düsseldorf, Germany) were prepared for surgery, and the head of the animal was fixed using a stereotaxic apparatus (Hugo Sachs Elektronik, Germany). A stainless steel guide cannula (22GA, 9 mm long, Plastic One, USA) was implanted into each lateral ventricle according to coordinates from the Paxinos and Watson atlas (coordinates: AP: −1.3 mm, L: ±2 mm, D: +3.6 mm according to bregma) [62]. The cannulae were permanently anchored to three stainless steel skull screws with dental acrylic (Duracryl, spofaDental a.s., Jicin, Czech Republic). STZ was administered through the implanted cannulae in a cumulative dose of 3 mg/kg over two injections on Days 2 and 4 (2 × 1.5 mg/kg, dissolved in 0.02 M citrate buffer pH 4.5) with separate injections into each lateral ventricle (0.75 mg/kg in 2 µL of vehicle) at a rate of 1 μL/min. The total dose of STZ was divided in order to reduce procedure-associated mortality [63]. During the infusion procedure, the animal under 2.5% isoflurane anaesthesia was held gently in the hand. The injections were performed using a microinfusion pump (Legato-100—Series Syringe Pump, KD SCIENTIFIC, Holliston, MA, USA) and a Hamilton syringe (10 μL) connected via polyethylene tube to an injection cannula (28GA, 11 mm long, Plastic One, Roanoke, VA, USA) which was placed into the guide cannula 2 mm below its tip. In order to avoid a backflow of the solution, the injection cannula was left inside the guide for an additional 60 s. The control groups received the same vehicle injection volumes by the same procedure. As the stability of STZ solution is maximum at pH close to 4, we used citrate buffer pH 4.5 as vehicle. After surgery, the animals were transferred to the worm room, where they stayed until regaining consciousness. Like other authors [64] using the open field test and the same dose of ICV injected STZ (3 mg/kg), we did not observe locomotor sensitization after ICV-STZ injections.

#### 4.2.3. Elevated Plus Maze Test (EPM)

Anxiety-related spatial memory and learning in the elevated plus maze (EPM) test was performed according to a procedure that we described previously [65] with some modifications. In order to increase fear/anxiety and the difference between the open and enclosed arms, the EPM test was made of plexiglass and consisted of two white, open arms 50 × 10 cm, and two opposite, black arms enclosed by 40 cm high walls. The arms were connected by a central 10 × 10 cm white square, and thus the maze formed a cross or a plus sign. The whole apparatus was elevated 50 cm above the floor. The EPM test was conducted between 8:00 and 12:00 A.M. in the test/re-test (after 1 h) paradigm. Rats were carried in their home cages to the experimental room, placed in the maze always in the same position (heading towards the open end of the maze) and allowed to explore the EPM for 5 min (test session). Then, the animals were carried back to their home cages for 60 min. After the 60 min interval, rats were returned to the experimental room and placed again in the EPM test for 5 min (re-test session). Recording was performed at baseline and on the 45th and 90th post-injection days (Figure 7) in the test/re-test paradigm.

In order to minimize bias, during the exposure of the rats to the EPM test, the behavioral responses of the individuals were recorded using a video camera, positioned approximately 200 cm over the center of the maze and connected to a video-tracking digitizing device (EthoVision XT, Noldus, Wageningen, The Netherlands). The EPM test was calibrated in the computer software, so the camera could create physical distance information from pixel-based data. After each rat testing, the whole apparatus was cleaned with a solution of 70% ethanol and left to dry for 5 min to avoid any influence of olfactory cues. Registered reactions included: time in the open/closed arms and the center, open/closed arms, and center entries. Then, a trained observer who was blinded to the experimental treatment of the rats scored behavioral responses.

#### 4.2.4. White and Illuminated Open Field Test (OF)

Recording was performed: at baseline and on the 45th and 90th post-injection days (Figure 7) according to the procedure that we have described previously [33]. The open area (100 × 100 × 60 cm) was made of white wood and was placed in a quiet and dark room adjacent to the housing room. Each arena surface was divided into 25 equal squares with white lines. A white-light lamp of 200 W was placed 75 cm over the center of the arena during testing (for the rodents, an illuminated and open compartment is considered to be more stressful than a dark enclosed one) [56]. The animals were gently transported in the home cages to the room and released into one of the corners of the arena (start corner). Each OF exposure lasted for 30 min, during which the animals were left undisturbed in the testing room. After the test, the arena was cleaned with water, alcohol, and water again.

In order to minimize bias, during the exposure of the rats to the OF test, the behavioral responses of the individuals were recorded using a video camera, positioned approximately 200 cm over the center of the maze and connected to a video-tracking digitizing device (EthoVision XT, Noldus, Wageningen, The Netherlands). Behavioral responses recorded included: exploration (measured as number of lines crossed), freezing time, time at periphery and center, center entries, rearing, grooming, defecation and miction (frequency—number of behaviors occurring) [66]. Then, a trained observer who was blinded to the experimental treatment of the rats scored behavioral responses.

#### 4.2.5. Test of Spatial Memory in Morris Water Maze (MWM)

Morris water maze (MWM) testing was determined at baseline and on the 45th and 90th post-injection days (after the first STZ or VEH injections (Figure 7). Spatial reference memory was tested first followed by spatial working memory according to the procedure that we described previously [61,67]. Briefly, MWM training with a visible platform was performed initially. If the rat did not locate the platform within 60 s, it was gently steered towards the platform by hand. After 5 min, a screening trial with a visible platform was performed in order to exclude rats with motivational or sensory-motor deficits. The spatial memory testing phase occurred over 8 days and was performed in two stages: reference memory testing with platform position remaining constant for all training sessions (3 days, 4 120 s trials per day and a probe test without a platform on the 4th day) and working memory testing with the platform position changed every day (4 days, 4 trials a day). Inter-trial interval (ITI) was 10 min. In order to minimize bias, during the exposure of the rats to the MWM test, the behavioral responses of the individuals were recorded using a video camera, positioned approximately 200 cm over the center of the maze and connected to a video-tracking (EthoVision XT, Noldus, Wageningen, The Netherlands). Three MWM parameters were measured with the use of a video-tracking digitizing device: latency to reach the platform (in most trials) or the critical annulus (CA—virtual contour of the removed platform) in the probe test, distance swum (path length), and the percentage of time spent in the critical quadrant (CQ) of the pool (where the platform was located). Then, a trained observer who was blinded to the experimental treatment of the rats scored behavioral responses.

#### 4.2.6. Immunohistochemistry

In order to determine activated microglia cells and unaggregated, oligomeric and fibrillar forms of beta Amyloid 42 and unaggregated beta Amyloid 40 (Aβ_1–42_ and Aβ_1–40_) we used immunofluorescence labeling on brain sections. Since numerous studies have shown that strong expression of the lysosomal protein CD68 can be used for activated microglia staining [68,69], we detected activated microglia cells (CD68^+^ cells) according to the method we described earlier [67]. Fixed brains were cut on a cryostat (CM1850, Leica Wetzlar, Germany) to obtain 20 µm thick sections. The sections were then placed in buffered saline (PBS) pH 7.4 containing 5% bovine serum albumin (BSA) and 0.3% Tris X for 2 h blocking. In the next step, the sections were rinsed 3 times for 5 min each with PBS solution. The tissue was then placed for 48 h in a primary antibody solution (mouse anti-rat, Sigma Aldrich, MAB1435, 1:250 for determination of CD68^+^ cells or the monoclonal mouse antibody anti-rat, Novusbio NBP2-13075, 1:300 for the detection of A*β* peptide 42 or 40) in PBS containing 5% BSA and 0.3% Tris-X. After another 3 washings, the sections were placed in a secondary antibody solution (goat anti-mouse, Abcam, Alexa Fluor 488, ab150113, 1:500 for both CD68^+^ microglia cells or A*β* peptide) in PBS for 2 h. After this procedure, the sections for labeling of CD68^+^ microglia cells were placed on gelatinized slides and covered with DPX solution to cover the slides with a coverslip. The prepared slides were photographed using a fluorescence microscope (Zeiss Axio Scope A1, Oberkochen Germany) compatible with the Axio Vision software. 2.10. The sections for detecting A*β* peptide deposits were then placed on gelatinized slides and covered with a fluoromount-G solution in dapi (Invitrogen, 00-4959-52, Waltham, MA, USA) to cover the slides with a coverslip. The prepared slides were photographed using a fluorescence microscope (DM1000 LED, Leica, Wetzlar, Germany) compatible with the LAS X program.

#### 4.2.7. Quantification of Labelled Cells

In order to minimize bias, counting of microglia cells and beta-amyloid deposits was performed by an observer who was blinded to the experimental treatment of the rats. CD68 positive (CD68^+^) cells and beta amyloid (A*β*) positive peptides were counted in both hemispheres of each tested structure. The borders of the brain structures were determined on the basis of the Paxinos and Watson atlas [62], outliner by a counter, and an area of the region was computed by the software. The image was adjusted through a threshold procedure such that neighboring areas with only nonspecific background staining would contain no positive signals. The number of microglia cells (CD68^+^) was counted with a calibrated frame over an area of 0.01 mm^2^ whereas the number of beta-amyloid (A*β*) peptide was counted with a calibrated frame over an area of 0.1 mm^2^. The density of CD68^+^ cells or A*β* peptide were computed (number per 1 mm^2^) in each tested area. When possible, the value of three separate counts including both brain hemispheres of an individual brain region for each animal (3 sections) was averaged and used as a single data point.

#### 4.2.8. Peripheral Blood Total Leukocyte and Leukocyte Subset Number

One hour after the last session of EPM, OF and Morris test, blood samples in a volume of 5 mL were collected by a cardiac puncture under 2.5% isoflurane (airflow: 0.5 l per min) anesthesia using an isoflurane pump (Bitmos OXY 6000, Bitmos GmbH, Düsseldorf, Germany). Then, rats were euthanized with Morbital (2 mL/kg), brain, spleen and thymus were harvested from STZ and VEH animals. The total leukocyte number and total and percentage of lymphocytes, granulocytes and monocytes were determined with the hematology analyzer (Horriba, Loos, France).

Lymphocyte populations and subpopulations were determined using cytometric analysis according to the method that we described previously [61,70] with some modifications. Peripheral blood mononuclear cells (PBMC) were obtained from heparinized whole blood by the Ficoll 400 (Pharmacia, Uppsala, Sweden) and Uropolinum (Polfa, Starogard, Poland) density centrifugation method. After the centrifugation (1113× *g*, 30 min at 4 °C), the isolated cells were collected with a Pasteur pipette, washed with phosphate-buffered saline (PBMC) suspension in RPMI-1640 with a 10% calf bovine serum, and seeded at a concentration of 4 × 10^6^ cells/mL and used in the flow cytometry analysis.

A three-color combination of fluorescent monoclonal antibodies was used in the blood and spleen lymphocyte study to identify T (CD3^+^), B (CD45RA^+^), NK (CD161a^+^) lymphocytes and T lymphocyte subsets of the T helper (CD3^+^CD4^+^CD8^−^) and T cytotoxic (CD3^+^CD4^−^CD8^+^) lymphocyte percentage. A total of 25 µL of the whole blood was added to 25 µL of IOTest CD3-FITC/CD45RA-PC7/CD161a-APC or CD3-FITC/CD4-PC7/CD8-APC (Beckman Coulter, Brea, CA, USA) according to the manufacturer’s instructions. Erythrocytes were lysed (Versalyse, Beckman Coulter), and then blood and spleen samples were mixed and incubated at room temperature for 20 min in darkness. After incubation, 25 μL of Fixative Solution (Beckman Coulter, Brea, CA, USA) and 700 μL of PBS was added to the separate sample. Samples were protected from light and stored at 4 °C until flow cytometry had been performed with a Cytomics FC 500 flow cytometer (Beckman Coulter, Brea, CA, USA) and MXP Software version 2.0. The percentages of lymphocyte population and T lymphocyte subsets were assessed during the assay, displaying forward- and side-scatter characteristics. Then, total numbers of the lymphocyte population and their subsets were calculated on respective counts of the total leukocyte number and a percentage of the T (CD3^+^), B (CD45RA^+^), NK (CD161a^+^), CD3^+^CD4^+^CD8^−^T, and CD3^+^CD4^−^CD8^+^T lymphocytes.

#### 4.2.9. Determination of Pro-Inflammatory Interleukin (IL)-6 and Anti-Inflammatory Interleukin (IL)-10

IL-6 and IL-10 concentrations in plasma were determined by enzyme-linked immunoassay (ELISA) using a commercially available kit (Rat-IL6 and Rat-IL-10 ELISA kits Thermo Scientific, Waltham, MA, USA) according to the manufacturer’s instructions and our previous study [35]. Briefly, 100 or 50 µL of standards or samples were dispensed into 96 wells coated with rat IL-6 and IL-10 antibody, respectively, and incubated for 2 h (IL-6) and 1 h (IL-10) at room temperature. After extensive washing, 100 µL of the biotinylated anti-IL-6 or anti-IL-10 were added to each well, and the plates were incubated for 1 h (IL-6 and IL-10) at room temperature. The wells were again washed 3 times, 100 µL of Streptavidin-HRP was added, and incubation was carried out for 30 min. 3,3′,5,5′-tetramethylbenzidine (TMB) (100 μL/well) was used as the chromogen for the colorimetric assay. The reaction was stopped after 10 min by adding 100 µL/well of stop solution, and the absorbance was determined using the DTX 880 Multimode Detector (Beckman Coulter, Brea, CA, USA) system set to 450 nm. Cytokine concentrations were calculated based on the standard curve generated by Beckman Coulter’s Biomek software version i5 program based on the absorbance of standard samples. The sensitivity of detection was 16 pg/mL for IL-6 and 3 pg/mL for IL-10. 

#### 4.2.10. Determination of Plasma Corticosterone (CORT) Concentration

Concomitantly with the immune parameters, in the same animals, the plasma corticosterone (CORT) concentrations were measured. Blood samples were collected via heart puncture under 2.5% isoflurane (airflow: 0.5 L per min) anesthesia using an isoflurane pump (Bitmos OXY 6000, Bitmos GmbH Düsseldorf, Germany) between 9:00 and 10:00 A.M., when COR levels are at their lowest [71]. The plasma CORT concentration was measured by radioimmunoassay using a commercially available kit (rCorticosterone (^125^I RIA KIT, isotop Institute of Isotopes Co, Ltd., Budapest, Hungary) and Wizard 1470 gamma counter (Pharmacia—LKB, Turku, Finland). The measures were made in a duplicate. The sensitivity of the assay was 0.01 ng/tube.

#### 4.2.11. Peripheral Blood Hematological Parameters

Peripheral blood hematological parameters—red blood cells (RBC), hemoglobin concentration (HGB), mean hemoglobin concentration in the red blood cell (MCHC), mean mass of the hemoglobin in the red blood cell (MCH), mean corpuscular volume (MCV), hematocrit (HCT), and red cell distribution width (RDW)—were determined in the heparinized whole blood with the hematology analyzer (Horriba, France).

#### 4.2.12. Data Analysis

The data are presented as the mean ± SD. The normality of the distribution of the variables was checked with Kołmogorow-Smirnov test and the homogeneity of the variances with a Levene test. As the outcome of Kołmogorow-Smirnov test indicated that all data, was not distributed normally, we used for further statistical evaluation of behavioral and neuroimmune parameters, non-parametric tests. Data were evaluated using the Kruskal–Wallis one-way ANOVA with factors AD-like induction (STZ_VEH), and response to novelty (HRs_LRs) for multiple comparisons, followed by the Mann–Whitney U test for comparisons between two groups. A *p* value lower than 0.05 was considered statistically significant.

## 5. Conclusions

The results suggest that mechanisms underlying individual differences in responsiveness to environmental factors, including stress reactivity and anxiety levels influence the progression of memory and anxiety disorders in streptozotocin-induced neuroinflammation used as a rat model of early pathophysiological changes in sporadic Alzheimer’s disease (sAD). Behavioral activity associated with memory and anxiety disorders, levels of neuroinflammation, and β-amyloid peptide accumulation in the brain are more pronounced in rats with high reactivity to novelty, suggesting that HRs have an elevated risk of developing sAD. These findings highlight that an individual’s behavioral characteristics should be considered as risk factors and predictors of sAD pathology and thus may help to develop future effective preventive and therapeutic approaches for this disease.

## Figures and Tables

**Figure 1 ijms-25-11562-f001:**
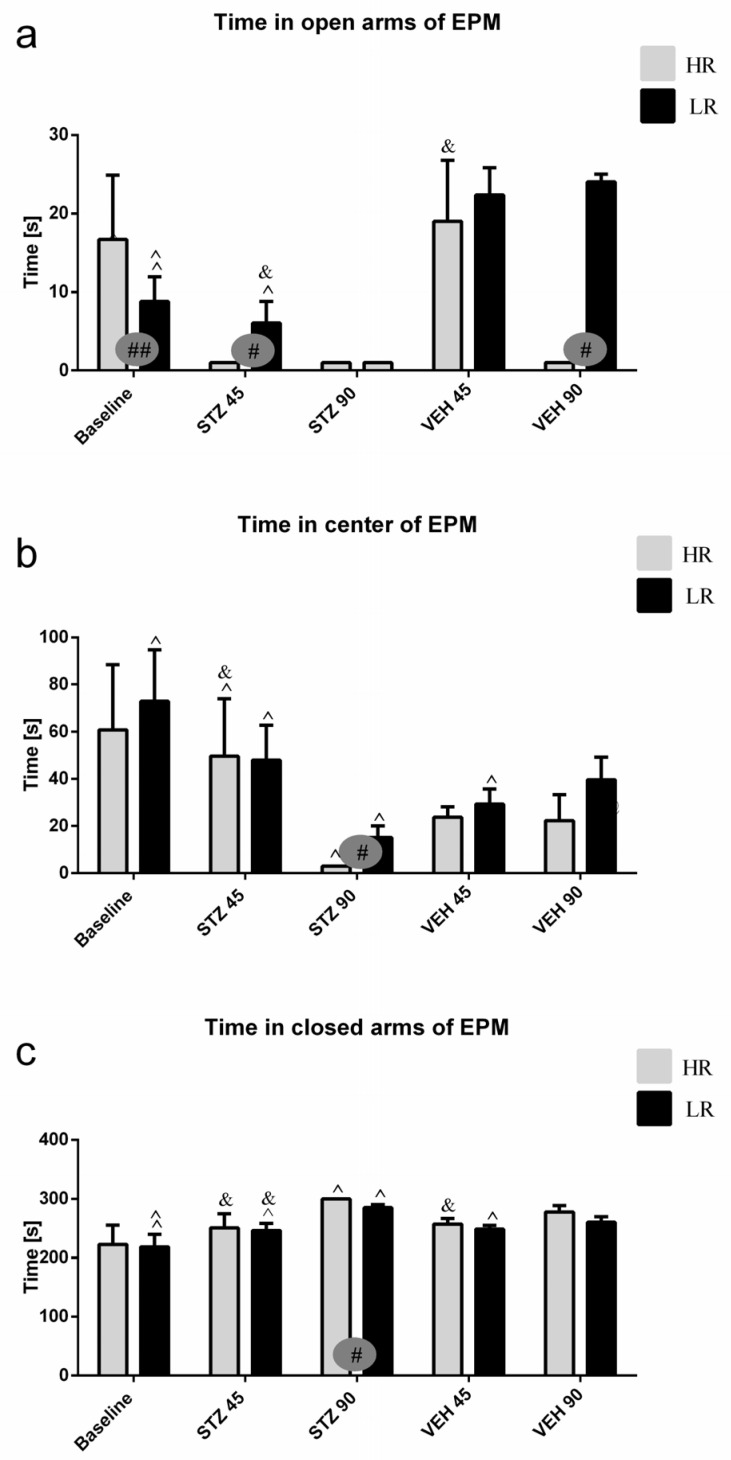
Behavioral activity associated with anxiety in the elevated plus maze (EPM) test measured as time spent in the open arms (**a**), time spent in the center (**b**), and time spent in the closed arms (**c**) of the maze in rats with high (HR) or low (LR) reactivity to novelty at baseline (before injections) and 45 and 90 days after intracerebroventricular injections of streptozotocin (STZ, n = 16) or citrate buffer (VEH, n = 14). Explanations: Data are presented as mean ± SD and were analyzed using a Mann–Whitney-*U* test; # in a painted circle—*p* ≤ 0.05, ## in a painted circle indicate significance of differences between HR and LR rats; ^—*p* ≤ 0.05, ^^—*p* ≤ 0.01 indicate significance of differences between test and re-test; &—*p* ≤ 0.05 indicates significance of differences between 45 day and 90 day after injection.

**Figure 2 ijms-25-11562-f002:**
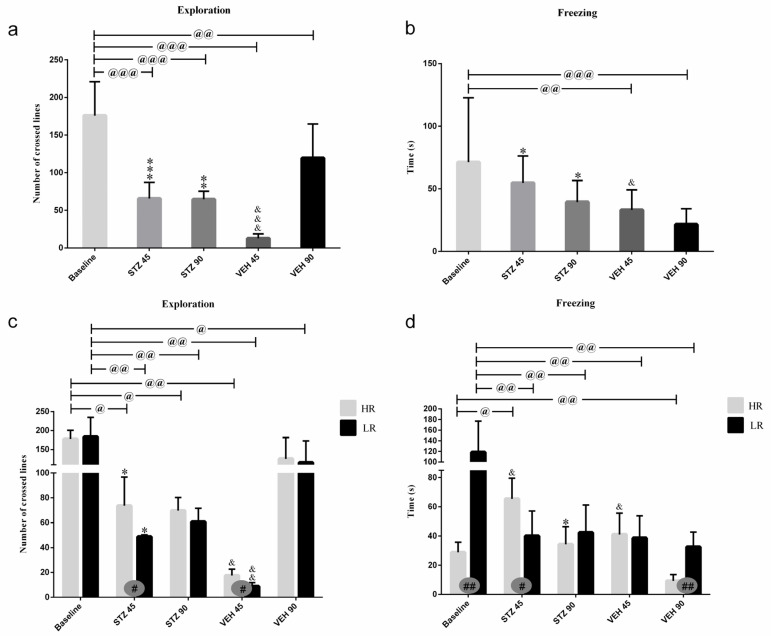
Behavioral activity associated with anxiety in the white and illuminated open field (OF) test measured as exploration (number of crossed lines) and time of freezing in rats non-divided (**a**,**b**) and divided into rats with high (HR) or low (LR) reactivity to novelty (**c**,**d**) at baseline (before injections) and 45 and 90 days after intracerebroventricular injections of streptozotocin (STZ, n = 16) or citrate buffer (VEH, n = 14). Explanations: Data are presented as mean ± SD and were analyzed using a Mann–Whitney-*U* test; # in a painted circle—*p* ≤ 0.05, ## in a painted circle indicate significance of differences between HR and LR rats; *—*p* ≤ 0.05, **—*p* ≤ 0.01, ***—*p* ≤ 0.001 indicate significance of differences between STZ and VEH at 45 or 90 days after injection; &—*p* ≤ 0.05, &&—*p* ≤ 0.01, &&&—*p* ≤ 0.001 indicate significance of differences between 45 days and 90 days after injection; @ in brackets above the bars—*p* ≤ 0.05, @@ in brackets above the bars—*p* ≤ 0.01, @@@ in brackets above the bars—*p* ≤ 0.001 indicate significance of differences vs. baseline.

**Figure 3 ijms-25-11562-f003:**
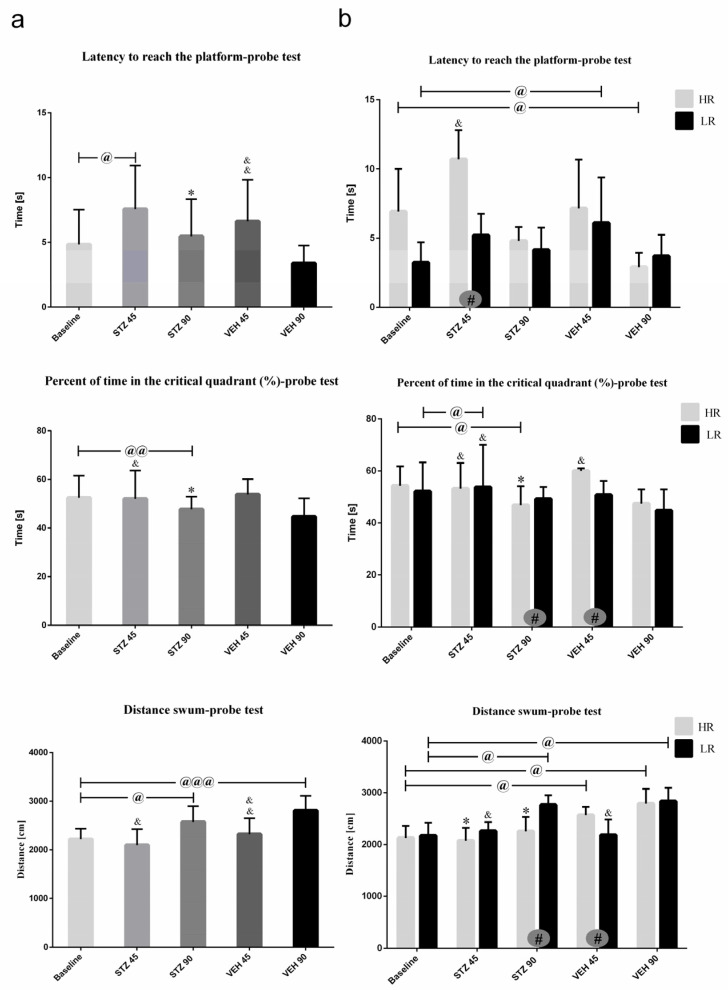
Behavioral activity associated with reference memory performance in the Morris water maze (MWM) test measured as latency to reach the platform, the percentage of time in the critical quadrant, and the total distance swum in the critical quadrant in a probe test in rats non-divided (**a**) and divided into rats with high (HR) or low (LR) reactivity to novelty (**b**) at baseline (before injections) and 45 and 90 days after intracerebroventricular injections of streptozotocin (STZ, n = 16) or citrate buffer (VEH, n = 14). Explanations: latency to reach the platform: latency to reach the critical quadrant in which the platform was located in the previous sessions of spatial learning in the MWM (Days 1–3); probe test: single trial without the platform (Day 4); data are presented as mean ± SD and were analyzed using a Mann–Whitney-U test; # in a painted circle—*p* ≤ 0.05 indicates significance of differences between HR and LR rats; *—*p* ≤ 0.05 indicates significance of differences between STZ and VEH at 45 or 90 days after injection; &—*p* ≤ 0.05, &&—*p* ≤ 0.01 indicate significance of differences between 45 day and 90 day after injection; @ in brackets above the bars—*p* ≤ 0.05, @@ in brackets above the bars—*p* ≤ 0.01 indicate significance of differences vs. baseline, @@@—*p* ≤ 0.001 indicate significance of differences vs. baseline.

**Figure 4 ijms-25-11562-f004:**
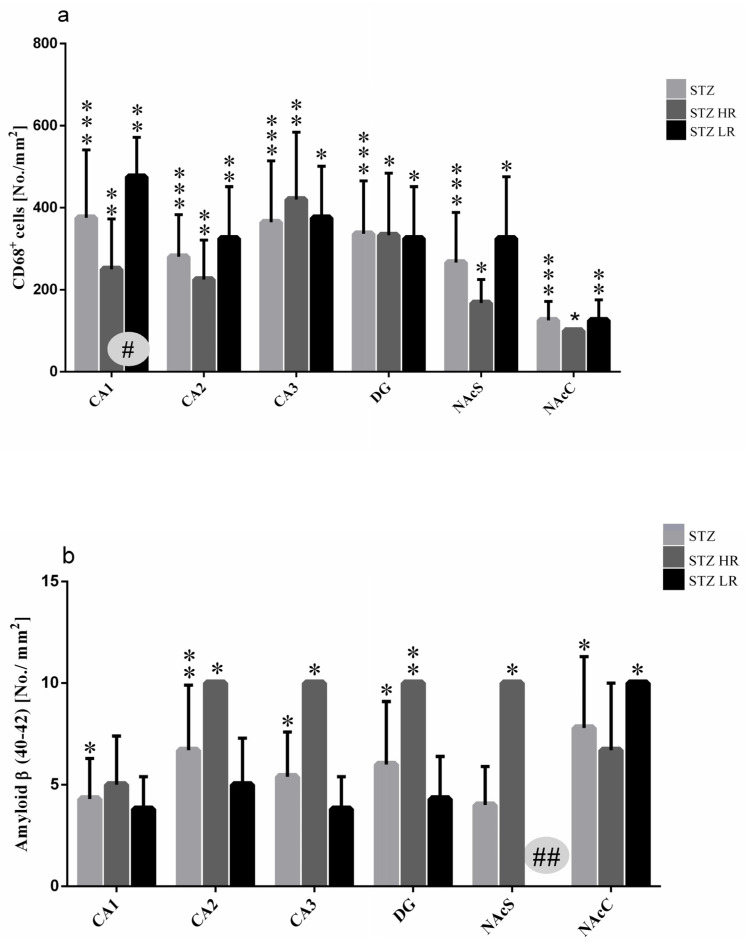
Number of activated microglia cells (CD68^+^ cells) (**a**) and β-amyloid counts (**b**) in the hippocampus (CA1, CA2, CA3, DG) and nucleus accumbens (NAcC, NAcS) in non-divided rats and divided into rats with high (HR) or low (LR) reactivity to novelty 90 days after intracerebroventricular injections of streptozotocin (STZ, n = 16). Explanations: Data are presented as mean ± SD and were analyzed using a Mann–Whitney-*U* test; # in a painted circle—*p* ≤ 0.05, ## in a painted circle *p* ≤ 0.01 indicate significance of differences between HR and LR rats; *—*p* ≤ 0.05, **—*p* ≤ 0.01, ***—*p* ≤ 0.001 indicate significance of differences between STZ and VEH 90 days after injection.

**Figure 5 ijms-25-11562-f005:**
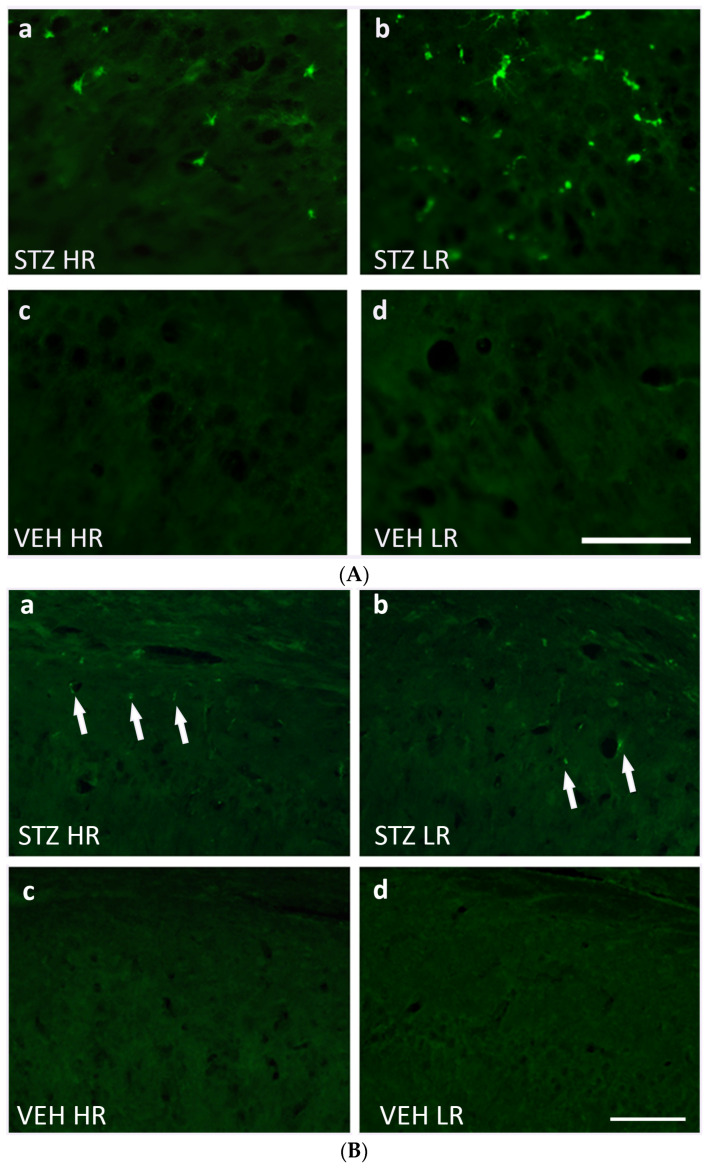
(**A**) Representative photomicrographs showing activated microglia cells (CD68^+^ cells) in the CA1 area of the hippocampus of STZ HR (**a**), STZ LR (**b**), VEH HR (**c**), and VEH LR rats (**d**) 90 days after intracerebroventricular injections of streptozotocin (STZ) or citrate buffer (VEH). Scale bar = 100 μm. (**B**) Representative photomicrographs showing amyloid beta (indicated by arrows) in the CA1 area of the hippocampus of STZ HR (**a**), STZ LR (**b**), VEH HR (**c**), and VEH LR rats (**d**) 90 days after intracerebroventricular injections of streptozotocin (STZ) or citrate buffer (VEH). Scale bar = 100 μm.

**Figure 6 ijms-25-11562-f006:**
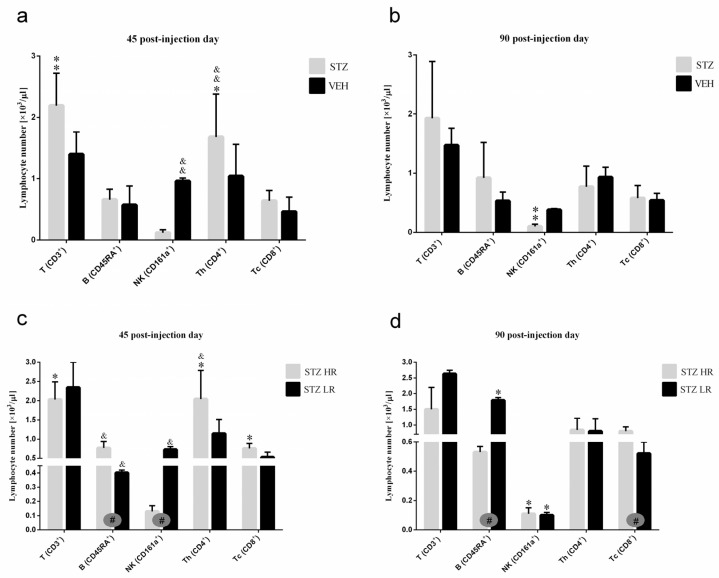
The total number of T, B, NK, TCD4^+^, and TCD8^+^ lymphocytes in the peripheral blood mononuclear cells analyzed by the flow cytometric method 45 and 90 days after intracerebroventricular injections of streptozotocin (STZ, n = 16) or citrate buffer (VEH, n = 14) in non-divided rats (**a**,**b**) and divided into rats with high (HR, n = 8) or low (LR, n = 8) reactivity to novelty after intracerebroventricular injections of STZ (**c**,**d**). Explanations: Data are presented as mean ± SD and were analyzed using a Mann–Whitney-*U* test; # in a painted circle—*p* ≤ 0.05 indicate significance of differences between HR and LR rats; *—*p* ≤ 0.05, **—*p* ≤ 0.01 indicate significance of differences between STZ and VEH 45 and 90 days after injection; &—*p* ≤ 0.05, &&—*p* ≤ 0.01 indicate significance of differences between 45 day and 90 day after injection.

**Figure 7 ijms-25-11562-f007:**
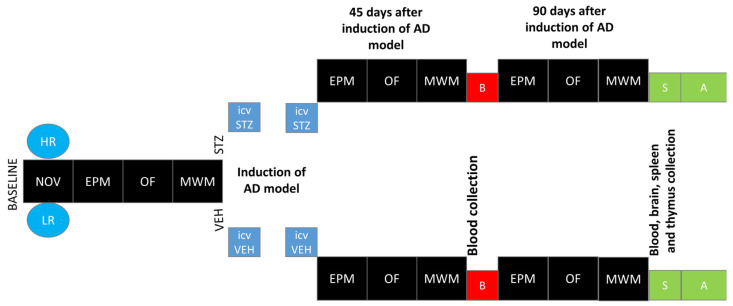
Scheme of experimental procedure and group assignments. Explanations: NOV—novelty test; EPM—elevated plus maze test: test and re-test after 1 h; OF—white and illuminated open-field test; MWM—Morris water maze test: phase of acquisition of reference memory (Day 1–Day 3), probe test (single trial without the platform (reference memory evaluation), working memory performance (Days 5–8); HR—high responders to novelty; LR—low responders to novelty; ICVSTZ—intracerebroventricular (ICV) injections of streptozotocin (STZ) (a total dose of 3 mg/kg b.w. divided into 2 injections to both lateral ventricles: 0.75 mg/kg STZ in 2 μL of citrate buffer per ventricle)—sAD model; ICVVEH—intracerebroventricular (ICV) injections of citrate buffer (VEH) (2 μL per ventricle); B—blood collection; S—1 h after the last MWM trial euthanasia and blood, brain, spleen, and thymus collection for further analysis (A).

**Table 1 ijms-25-11562-t001:** Behavioral activity in the white and illuminated open field (OF) test measured as time spent in the periphery and center, entries to the center, rearing, grooming, miction, and defecation in rats non-divided and divided into rats with high (HR) or low (LR) reactivity to novelty at baseline (before injections) and 45 and 90 days after intracerebroventricular injections of streptozotocin (STZ, n = 16) or citrate buffer (VEH, n = 14).

Parameter/Group		Rearing (No.)	Grooming (No.)	Time at Periphery (s)	Time at the Center (s)	Center Entries (No.)	Miction (No.)	Defecation (No.)
Baseline Mean ± SD	Non-divided	14.39 ± 6.66	2.70 ± 1.11	1791.7 ± 7.70	8.3 ± 7.70	4.67 ± 4.32	5.69 ± 2.55	4.75 ± 2.18 ^##^
	HR	14.71 ± 6.47	2.71 ± 0.76	1798.67 ± 0.58 ^#^	1.33 ± 0.58 ^#^	1.33 ± 0.58 ^#^	4.14 ± 1.68	3.25 ± 0.5
	LR	15.91 ± 7.22	2.82 ± 1.08	1784.73 ± 1.55	15.27 ± 1.55	8. ± 3.61	6.36 ± 3.04	5.67 ± 2.35
STZ 45 Mean ± SD	Non-divided	10.25 ± 3.23 **^&&&^	2.43 ± 0.79	1800 ± 0 ^@@^	0 ± 0 ^@@^	0 ± 0 ^@@^	1.63 ± 0.52 *^@@@^	3.63 ± 1.60 *
	HR	10 ± 1 ^$&^	3 ± 0 ^$#&^	1800 ± 0 ^@^	0 ± 0 ^@^	0 ± 0 ^@^	1.66 ± 0.58 ^@^	4.33 ± 1.15 ^$^
	LR	10.75 ± 4.92 ^&^	1.67 ± 0.58 ^$^	1800 ± 0 ^@^	0 ± 0 ^@^	0 ± 0 ^@^	1.75 ± 0.5 ^$;@^	3.75 ± 1.5
STZ 90 Mean ± SD	Non-divided	3 ± 1.41 ^@@@^	1.71 ± 0.49 ^@^	1800 ± 0 ^@@^	0 ± 0 ^@@^	0 ± 0 ^@@^	1.17 ± 0.41 ^@@@^	4 ± 1.26 *
	HR	3 ± 1.41 ^@^	1.67 ± 0.58	1800 ± 0 ^@^	0 ± 0 ^@^	0 ± 0 ^@^	1.33 ± 0.58 ^@^	3 ± 0 ^#^
	LR	3 ± 1.41 ^@@^	1.75 ± 0.5	1800 ± 0 ^@^	0 ± 0 ^@^	0 ± 0 ^@^	1 ± 0 ^$@@^	5 ± 1
VEH 45 Mean ± SD	Non-divided	6.2 ± 1.87 ^&&&@@@^	3.13 ± 2.36	1800 ± 0 ^@@@^	0 ± 0 ^@@@^	0 ± 0 ^@@@^	1.08 ± 0.29 ^@@@^	2.13 ± 1.55 ^@@^
	HR	4.5 ± 0.58 ^#&@@^	1 ± 0 ^#;@^	1800 ± 0 ^@^	0 ± 0 ^@^	0 ± 0 ^@^	1.33 ± 0.58 ^@^	1 ± 0 ^#;@^
	LR	8 ± 1.73 ^&@^	4.4 ± 2.07 ^&^	1800 ± 0 ^@^	0 ± 0 ^@^	0 ± 0 ^@^	1 ± 0 ^&&;@@^	3.25 ± 1.5
VEH 90 Mean ± SD	Non-divided	2.71 ± 0.95 ^@@@^	1.7 ± 0.67 ^@^	1800 ± 0 ^@@@^	0 ± 0 ^@@@^	0 ± 0 ^@@@^	0.67 ± 0.65 ^@@@^	2.55 ± 1.04 ^@@^
	HR	3 ± 0 ^@^	1.33 ± 0.58 ^@^	1800 ± 0 ^@^	0 ± 0 ^@^	0 ± 0 ^@^	1.25 ± 0.5 ^##@@^	2.25 ± 1.26
	LR	2 ± 1 ^@@^	1.8 ± 0.84	1800 ± 0 ^@^	0 ± 0 ^@^	0 ± 0 ^@^	0 ± 0 ^@@^	3 ± 1.15 ^@^

Explanations: Data are presented as mean ± SD and were analyzed using a Mann–Whitney-U test; #—*p* ≤ 0.05, ##—*p* ≤ 0.01 indicate significance of differences between HR and LR within the STZ or VEH animals; *—*p* ≤ 0.05, **—*p* ≤ 0.01 indicate significance of differences between STZ and VEH at 45 or 90 days after injection; &—*p* ≤ 0.05, &&—*p* ≤ 0.01, &&&—*p* ≤ 0.001 indicate significance of differences between 45 days and 90 days after injection; @—*p* ≤ 0.05, @@—*p* ≤ 0.01, @@@—*p* ≤ 0.001 indicate significance of differences vs. baseline; $—*p* ≤ 0.05, indicates significance of differences between STZHR and VEHHR or STZLR and VEHLR.

**Table 2 ijms-25-11562-t002:** Behavioral activity associated with working memory during Trial 1–4 of one day (a) and reference memory over 1–3 consecutive days (b) in the Morris water maze (MWM) test measured as latency to reach the platform and total distance swum in the critical quadrant in rats non-divided and divided into rats with high (HR) or low (LR) reactivity to novelty at baseline (before injections) and 45 and 90 days after intracerebroventricular injections of streptozotocin (STZ, n = 16) or citrate buffer (VEH, n = 14).

**(a)**	**Latency to Reach the Platform (s)**				
	**Group**	**Trial 1**	**Trial 2**	**Trial 3**	**Trial 4**
	Baseline	92.02 ± 30.26	7.51 ± 3.9 ^^^^^	5.28 ± 2.6 ^%%^	4.78 ± 2.1
	STZ 45	100.43 ± 19.78 ***^&&&^	86.08 ± 27.12 ***^&&&@@^^	78.09 ± 35.93 ***^&&&@@@^	82.98 ± 29.89 ***^&&&@@@^
	STZ 90	25.43 ± 9.84 ***^@@@^	19.73 ± 10.68 ***^@@@^^	22.03 ± 14.32 ***^@@@^	20.18 ± 13.75 ***^@@@^
	VEH 45	36.75 ± 11.08 ^&&&@@@^	6.68 ± 3.37 ^&&&^^^^	3.41 ± 1.37 ^@@%%%^	2.88 ± 1.26 ^@@@!^
	VEH 90	10.89 ± 6.41 ^@@@^	3.73 ± 1.43 ^@@@^^^^	3.28 ± 1.35 ^@@@^	2.83 ± 1.05 ^@@@^
	Baseline HR	90.68 ± 28.26	6.08 ± 2.58 ^#^^^^	4.95 ± 2.38 ^%^	5.08 ± 2.41
	Baseline LR	93.37 ± 32.68	8.95 ± 4.50 ^^^^^	5.6 ± 2.8 ^%%^	4.48 ± 1.73
	STZ 45 HR	99.63 ± 20.62 ^$$$&&&^	90.38 ± 27.43 ^$$$&&&@@@^	81.57 ± 37.53 ^$$$&&&@@@^	87.4 ± 25.53 ^$$$&&&@@@^
	STZ 45 LR	101.22 ± 19.77 ^$$$&&&^	81.78 ± 27.30 ^$$$&&&@@@^^	74.62 ± 35.56 ^$$$&&@@@^	78.55 ± 34.25 ^$$$&&@@@^
	STZ 90 HR	22.35 ± 8.87 ^@@@^	12.33 ± 5.4 ^$$$###@@@^^^	10.95 ± 5.09 ^$$$###@@@^	7.9 ± 2.79 ^$$$#@@^
	STZ 90 LR	28.52 ± 10.15 ^$$$@@@^	27.42 ± 9.06 ^$$$@@@^	33.5 ± 10.77 ^$$$@@@^	32.47 ± 7.66 ^@@@^
	VEH 45 HR	39.88 ± 10.89 ^&&&@@@^	8.45 ± 3.7 ^&&#@^^^^	3.64 ± 1.76 ^%%^	3.32 ± 1.57 ^@@@^
	VEH 45 LR	33.60 ± 10.80 ^&&&@@@^	4.9 ± 1.8 ^@@^^^^	3.20 ± 0.87 ^@@%^	2.45 ± 0.68 ^@@@!^
	VEH 90 HR	16.15 ± 4.35 ^###@@@^	3.87 ± 1.65 ^@@^^^	3.37 ± 1.24 ^@^	2.53 ± 0.81 ^@^
	VEH 90 LR	5.62 ± 2.56 ^@@@^	3.6 ± 1.22 ^@@@^^	3.2 ± 1.49 ^@@^	3.12 ± 1.22 ^@^
**(b)**	**Latency to Reach the Platform (s)**				**Total Distance Swum (cm)**
	**Group**	**Day 1**	**Day 2**	**Day 3**	**Day 1**
	Baseline	97.32 ± 44.71	26.58 ± 12.90 ^^^^^	20.39 ± 12.35 ^^^^%^	625.23 ± 293.05
	STZ 45	42.23 ± 17.02 ***^&@@^	10.97 ± 3.38 *^@@@^^^	11.09 ± 7.93 **^@^^^	715.74 ± 324.31 ***^&^
	STZ 90	23.79 ± 8.89 ***^@@@^	17.53 ± 8.56 ***^@^	13.16 ± 9.60 **^^^	428.65 ± 114.29 ***^@^
	VEH 45	8.52 ± 1.81 ^&@@@^	6.54 ± 4.83 ^@@@^^	4.64 ± 2.32 ^@@@^^^	255.83 ± 141.97 ^@@@^
	VEH 90	5.75 ± 2.28 ^@@@^	4.36 ± 1.99 ^@@@^	3.69 ± 1.24 ^@@@^^^	148.40 ± 41.49 ^@@@^
	Baseline HR	100.98 ± 44.66	22.7 ± 9.5 ^^^^	17.15 ± 5.97 ^^^^^	643.96 ± 280.13
	Baseline LR	86.38 ± 27.86	26.85 ± 10.63 ^^^^^	17.2 ± 8.27 ^^^^%^	578.88 ± 280.76
	STZ 45 HR	28.87 ± 10.25 ^$#@^	13.77 ± 3.7 ^$#^^	17.92 ± 7.98 ^$#^	426.02 ± 158.81 ^#^
	STZ 45 LR	51.2 ± 16.99 ^$&^	8.9 ± 0.35 ^@@^^	6.02 ± 1.52 ^&@^%^	993.47 ± 251.72 ^$&@^
	STZ 90 HR	30.62 ± 0.08 ^$#@^	18.38 ± 10.13 ^$^^	13.63 ± 6 ^$#^^	504.45 ± 66.46 ^$^
	STZ 90 LR	13.2 ± 2.85 ^$@@^	10.72 ± 2.33 ^$@@^	3.97 ± 0.21 ^@@^%^	374.04 ± 124.45 ^$^
	VEH 45 HR	9.66 ± 0.65 ^#&&@@^	4.28 ± 2.04 ^@@^^	3.85 ± 2.23 ^@@^^	339.36 ± 157.47 ^&##@^
	VEH 45 LR	7.75 ± 1.96 ^@@@^	8.05 ± 5.72 ^@@^	5.17 ± 2.43 ^@@@^	158.81 ± 28.66 ^@@@^
	VEH 90 HR	5.88 ± 2.15 ^@@^	4.53 ± 2.17 ^@@^	3.59 ± 1.31 ^@@^	152.67 ± 46.93 ^@@^
	VEH 90 LR	5.66 ± 2.54 ^@@@^	4.24 ± 2.02 ^@@@^	3.76 ± 1.28 ^@@@^	145.96 ± 41.82 ^@@@^

Explanations: Data are presented as mean ± SD and were analyzed using a Mann–Whitney-U test; #—*p* ≤ 0.05, ##—*p* ≤ 0.01, ###—*p* ≤ 0.001 indicate significance of differences between HR and LR within the STZ or VEH animals; *—*p* ≤ 0.05, **—*p* ≤ 0.01,*** *p* ≤ 0.001 indicate significance of differences between STZ and VEH 45 or 90 days after injection; &—*p* ≤ 0.05, &&—*p* ≤ 0.01, &&& *p* ≤ 0.001 indicate significance of differences between 45 day and 90 day after injection; @—*p* ≤ 0.05, @@—*p* ≤ 0.01, @@@—*p* ≤ 0.001 indicate significance of differences vs baseline; $—*p* ≤ 0.05, $$$—*p* ≤ 0.001 indicate significance of differences between STZHR and VEHHR or STZLR and VEHLR; ^—*p* ≤ 0.05, ^^—*p* ≤ 0.01, ^^^—*p* ≤ 0.001 indicate significance of differences between Trial 1; %—*p* ≤ 0.05, %%—*p* ≤ 0.01, %%%—*p* ≤ 0.001 indicate significance of differences between Trial 2; !—*p* ≤ 0.05 indicates significance of differences between Trial 3.

**Table 3 ijms-25-11562-t003:** Number of activated microglia cells (CD68^+^ cells) (a) and β-amyloid counts (b) in brain structures in rats non-divided and divided into rats with high (HR) or low (LR) reactivity to novelty 90 days after intracerebroventricular injections of streptozotocin (STZ, n = 16) or citrate buffer (VEH, n = 14).

**(a)**	**Microglia Cells (CD68^+^) No./mm^2^**						
	**Group/Structure**	**STZ** **Mean ± SD**	**VEH** **Mean ± SD**	**STZ HR** **Mean ± SD**	**STZ LR** **Mean ± SD**	**VEH HR** **Mean ± SD**	**VEH LR** **Mean ± SD**
	Prefrontal cortex	117 ± 41 ***	0 ± 0	133 ± 58 ^$^	100 ± 0 ^$^	0 ± 0	0 ± 0
	Corpus callosum	1018 ± 268 ***	280 ± 131	1000 ± 322 ^$$^	1000 ± 23 ^$$^	300 ± 115	267 ± 170
	Caudate putamen	463 ± 207 **	100 ± 0	325 ± 50 ^$^	533 ± 208 ^$^	100 ± 0	100 ± 0
	Medial septal nucleus	473 ± 142 ***	0 ± 0	400 ± 155 ^$$^	550 ± 58 ^$$^	0 ± 0	0 ± 0
	Lateral preoptic area	200 ± 87 ***	0 ± 0	175 ± 50 ^$$^	200 ± 115 ^$^	0 ± 0	0 ± 0
	Medial preoptic nucleus	244 ± 124 ***	0 ± 0	225 ± 150 ^$^	250 ± 129 ^$^	0 ± 0	0 ± 0
	Supraoptic nucleus	189 ± 93 *	100 ± 0	150 ± 58	200 ± 115	100 ± 0	100 ± 0
	Arcuate hypothalamic nucleus	464 ± 225 **	100 ± 65	767 ± 58	300 ± 0	0 ± 0	233 ± 115
	Dorsomedial hypothalamic nucleus	220 ± 103 ***	0 ± 0	300 ± 71 ^#$$^	125 ± 50 ^$^	0 ± 0	0 ± 0
	Lateral hypothalamus	220 ± 79 ***	0 ± 0	250 ± 58 ^$$^	133 ± 58 ^$^	0 ± 0	0 ± 0
	Paraventricular hypothalamic nucleus	163 ± 74 ***	0 ± 0	160 ± 89 ^$$^	167 ± 58 ^$^	0 ± 0	0 ± 0
	Ventromedial hypothalamic nucleus	113 ± 35 ***	0 ± 0	133 ± 58 ^$^	100 ± 0 ^$^	0 ± 0	0 ± 0
**(b)**	**Group/Structure**	**Amyloid β (40–42)/No./mm^2^**					
		**STZ** **Mean ± SD**	**VEH** **Mean ± SD**	**STZ HR** **Mean ± SD**	**STZ LR** **Mean ± SD**	**VEH HR** **Mean ± SD**	**VEH LR** **Mean ± SD**
	Prefrontal cortex	2.5 ± 2.25	0 ± 0	5 ± 2.75	0 ± 0	0 ± 0	0 ± 0
	Lateral preoptic area	8.9 ± 1.65 **	0 ± 0	8.3 ± 2.05 ^$^	1 ± 0 ^$^	0 ± 0	0 ± 0
	Medial preoptic nucleus	5 ± 2.6	0 ± 0	6.67 ± 2.6	3.33 ± 2.6	0 ± 0	0 ± 0

Explanations: Data are presented as mean ± SD and were analyzed using Mann–Whitney-*U* test; #—*p* ≤ 0.05 indicates significance of differences between HR and LR within the STZ animals; *—*p* ≤ 0.05, **—*p* ≤ 0.01, ***—*p* ≤ 0.001 indicate significance of differences between STZ and VEH at 90 days after injection; $—*p* ≤ 0.05, $$—*p* ≤ 0.01 indicate significance of differences between STZHR and VEHHR or STZLR and VEHLR.

## Data Availability

Data is contained within the article and Appendix A.

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
