# Peer review of "High Behavioral Reactivity to Novelty as a Susceptibility Factor for Memory and Anxiety Disorders in Streptozotocin-Induced Neuroinflammation as a Rat Model of Alzheimer’s Disease"

_ijms, 2024, doi:10.3390/ijms252111562_

Round 1
Reviewer 1 Report
Comments and Suggestions for Authors
Within the manuscript the experimental procedures are described, but it lacks detailed information on the randomization and blinding methods used during the trials. The authors are encouraged to add this information for minimizing bias.
The interpretation of the results could be more nuanced. For instance, the authors attribute the differences in neuroinflammation and cognitive performance solely to behavioral reactivity without considering other potential biological or environmental factors that could influence these outcomes. A discussion on alternative explanations or confounding factors would enhance the robustness of the conclusions drawn.
The manuscript does not adequately address the limitations of the study such as sample size, the generalizability of the findings, and the limitations inherent in using animal models to study human diseases.
The discussion section could be completed with the contextualization of the findings within the existing literature. When referencing previous studies, this should compare or contrast the results with those of other relevant research, which could help to highlight the significance of the findings.
The conclusion section could be strengthened by suggesting specific future research directions based on the findings or potential therapeutic approaches for Alzheimer's disease. Also, the authors should check the text formatting.
The paragraph with the lines 600-603 should be aligned using Justify style.
"The manuscript describes the experimental procedure, but it does not provide information regarding the randomization of experimental groups and blinding during trials. This is an invitation to the authors to complete the manuscript with this information in view of minimizing bias.
In the discussion section, the authors relate that behavioral reactivity is responsible in general to neuroinflammation and cognitive performance changes, without mentioning about possible biological or environmental reasons for such an effect. I encourage them to add a discussion paragraph regarding the alternative interpretations or other factors to increase the solidity of the conclusions.
Also, by referencing previous studies, the authors should compare the obtained results with other litterature research, that might help to highlight the importance of the findings.
The conclusion section could be improved by suggesting some future research or potential therapeutic strategies for Alzheimer's disease.
The authors should check text formating, e.g. paragraph with the lines 600-603."
Comments on the Quality of English LanguageThe English quality of the manuscript is good
Reviewer 2 Report
Comments and Suggestions for Authors
The article is well written and presents an interesting and relevant topic. Its main novelty is that behavioural activity associated with memory and anxiety disorders, levels of neuroinflammation and the accumulation of β-amyloid peptide in the brain after ICV-STZ injections have been shown to be more pronounced in rats with high reactivity to novelty, suggesting that HRs have a high risk of developing sAD. However, there are some points that need to be improved before it can be accepted:
1 - the statistics in the graphs are out of place and very confusing. Therefore, I suggest putting them in the form of lines between columns.
2 - the statistics should also be on the bottom of the graph, so that it's not too complicated to know which statistical test has been used.
3 - The article is too dense, particularly in terms of the results, so some tables should be added to supplement the material. In particular, tables 1 and 5.
4 - Figure 5 is also not very clear, please improve the image.
5 - The conclusions could be improved, particularly the realisation of the clinical importance of the study.
Reviewer 3 Report
Comments and Suggestions for Authors
This a very well presented and interesting manuscript centered on the conclusions that were extracted from an animal study in rats, attempting to evaluate the development of memory and anxiety disorders in sporadic Alzheimer’s disease. The overall protocol is described with details and does not include any points that merit correction, as the design of the research is adequate. The quality of presentation is higher than average, the methods are adequately described and the results are clearly presented.
This research is centered on the investigation of the
possible role of behavioral characteristics of rats, regarding their
influence on the progression of sporadic AD and anxiety.
I consider that this topic is both original and relevant to the field, attempting to address a specific gap in the field. This is the case because it is not well known whether individual differences in responsiveness to environmental factors, including stress reactivity and anxiety levels, should be considered as risk factors for development of memory and anxiety disorders in sporadic Alzheimer’s disease.
The original-novel point of this study, compared with other published material, is the investigation of whether behavioral characteristics of the HR and LR rats influence the progression of sAD.
The methodology of the authors seems to be adequate to support the presented results. Based on that, I do not consider that further controls are necessary.
The extracted conclusions are consistent with the evidence and arguments presented and they address the main question of the authors. This is true to to the sufficient number of rats that were included in the survey and because of the statistical analysis that accompanies the survey. The reported p-values are of statistical significance.
All the references are appropriate and presented in the correct style.
No further comments on the tables and figures are applicable, from my side.
The only issue is that the title does not declare that your results are based in an animal model of rats.
Comments on the Quality of English LanguageThe quality of English language is satisfactory, although minor editting is required.
Round 2
Reviewer 1 Report
Comments and Suggestions for Authors
The authors improved well the manuscript following my recommendations. I consider now suitable to be accepted in current form.
Reviewer 2 Report
Comments and Suggestions for Authors
This article was revised appropriately.
I recommend accept